# Usefulness of Preoperative High Systemic Immune-Inflammation Index as a Prognostic Biomarker in Patients Who Undergo Radical Cystectomy for Bladder Cancer: Multicenter Analysis

**DOI:** 10.3390/diagnostics11122194

**Published:** 2021-11-25

**Authors:** Shimpei Yamashita, Yuya Iwahashi, Haruka Miyai, Nagahide Matsumura, Keizo Hagino, Kazuro Kikkawa, Yasuo Kohjimoto, Isao Hara

**Affiliations:** 1Department of Urology, Wakayama Medical University, 811-1 Kimiidera, Wakayama 641-0012, Japan; 0911yuya@wakayama-med.ac.jp (Y.I.); kzro@wakayama-med.ac.jp (K.K.); ykohji@wakayama-med.ac.jp (Y.K.); hara@wakayama-med.ac.jp (I.H.); 2Department of Urology, Kinan Hospital, 46-70 Shinjyo, Tanabe, Wakayama 646-8588, Japan; najahide@gmail.com; 3Department of Urology, Rinku General Medical Center, 2-23 Rinkuoraikita, Izumisano, Osaka 598-8577, Japan; haruka_19890330@yahoo.co.jp (H.M.); k-hagino@rgmc.izumisano.osaka.jp (K.H.); 4Department of Urology, Kishiwada City Hospital, 1001 Gakuhara, Kishiwada, Osaka 596-8501, Japan

**Keywords:** bladder cancer, cystectomy, prognosis, inflammation, systemic immune-inflammation index

## Abstract

Evidence of the prognostic value of pretreatment systemic immune-inflammation index (SII) after radical cystectomy (RC) for bladder cancer is limited. This study aims to assess the association between preoperative SII and prognosis after RC for bladder cancer. In this multicenter retrospective study, we calculated preoperative SII as well as the neutrophil-lymphocyte ratio (NLR) and platelet-lymphocyte ratio (PLR) in 237 patients who underwent RC for bladder cancer between March 2009 and March 2018. Patients were classified into high SII and low SII groups by using the optimal cutoff value (438 × 10^9^/L) based on receiver operating characteristic curve analysis for cancer-specific death. We compared cancer-specific survival (CSS) and overall survival (OS) between the two groups. To evaluate the prognostic impact of preoperative SII, we also performed Cox proportional regression analyses for CSS and OS. Of 237 patients, 127 patients were classified into the high SII group and 110 patients into the low SII group. During the follow-up period, 70 patients died of bladder cancer (30%) and 21 patients died from other causes (9%). Patients with high SII had significantly lower rates of CSS and OS than those with low SII (*p* < 0.01 and *p* < 0.01, respectively). Multivariable Cox proportional hazard analysis showed that high SII was independently associated with poor CSS (*p* = 0.01) and poor OS (*p* < 0.01). In conclusion, high SII could be an independent significant predictor of poor prognosis after RC in patients with bladder cancer.

## 1. Introduction

Bladder cancer is the ninth most common cancer worldwide [1]. Radical cystectomy (RC) remains the standard treatment for localized muscle-invasive bladder cancer, [2] and is also indicated for patients with high-risk non-muscle invasive carcinoma or bacillus Calmette–Guerin refractory carcinoma in situ [3,4]. However, prognosis in patients who undergo RC is poor and their five-year overall survival (OS) rate after RC is around 50% [5]. Reliable biomarkers to predict prognosis in patients who undergo RC are therefore desired, and several possible predictors have been examined.

In patients with malignancies, systemic inflammatory response to tumor cells is known to be a representative prognostic factor and reflects circulation of inflammation cells. Several inflammatory cell markers, including neutrophil-lymphocyte (NLR) and platelet-lymphocyte ratio (PLR), have been used to predict prognosis in patients with various cancers, including bladder cancer [6]. Systemic immune-inflammation index (SII), which is calculated using peripheral lymphocyte, neutrophil and platelet counts, is a novel and useful biomarker of prognosis in patients with malignancies such as lung, breast, gastrointestinal, gynecological and hepatocellular cancers [7,8,9,10].

NLR and PLR have been shown to be useful for predicting prognosis in patients who undergo RC for bladder cancer [6,11,12]. Conversely, the impact of preoperative SII on prognosis after RC in patients with bladder cancer is less well understood. The role of SII as a reliable biomarker to predict survival after RC requires further research results. This multi-institutional study aims to investigate the association between preoperative SII and prognosis in patients after RC for bladder cancer.

## 2. Materials and Methods

### 2.1. Patients

We retrospectively reviewed the medical records of consecutive patients between March 2009 and March 2018 who underwent RC for bladder cancer at our three institutions (Wakayama Medical University Hospital, Kinan Hospital and Rinku General Medical Center). Lymph node dissection (common iliac bifurcation, internal iliac, presacral, obturator fossa and external iliac nodes) was also generally performed in these institutions in addition to cystectomy. Patients with clinical T2-T4a N0M0 bladder cancer were counseled to receive cisplatin-based neoadjuvant chemotherapy (NAC). Adjuvant chemotherapy was not generally offered, regardless of pathological diagnosis. Patients received regular post-operative follow-up with computed tomography, urine and blood examination every three months for two years, then every six months for the next three years, and every twelve months thereafter. Excluded from the present study were patients whose preoperative blood examination data were unavailable and who did not receive post-operative follow-up at our institutions. Of 239 candidates, 237 patients were finally enrolled in this study. This multi-center retrospective study was conducted in accordance with the Declaration of Helsinki and approved by the Wakayama Medical University Institutional Review Board (approval number 3008).

### 2.2. Data Collection

Patient demographic data at the time of operation, including age, gender, body mass index (BMI), Eastern Cooperative Oncology Group performance status (ECOG PS), Charlson Comorbidity Index (CCI) and laboratory test results were collected retrospectively from medical records. Laboratory test results included neutrophil count (/L), lymphocyte count (/L) and platelet count (/L). SII was calculated as platelet count × neutrophil count/lymphocyte count. NLR and PLR were calculated as neutrophil count/lymphocyte count and platelet count/lymphocyte count, respectively. Information about NAC, cystectomy approach (open, laparoscopic or robotic), the type of urinary diversion (cutaneous ureterostomy, ileal conduit or neobladder) and histopathological data of resected specimens (pathological diagnosis, pathological T stage, presence of pathological lymph node metastasis and presence of concurrent CIS) were also collected.

### 2.3. Statistical Analysis

JMP Pro 14 (SAS Institute Inc., Cary, NC, USA) was used for all statistical analyses. We constructed receiver operating characteristic (ROC) curves to calculate the optimal cutoff values of NLR, PLR and SII for predicting cancer-specific death. Patients were classified into high and low NLR groups, high and low PLR groups and high and low SII groups by using each cutoff value, respectively. Cancer-specific survival (CSS) rate and OS rate were determined by the Kaplan–Meier method. Comparisons of CSS and OS between groups were performed using log rank tests. Comparison of patient demographics between two groups were performed using chi-square tests, Fisher’s exact tests or Mann–Whitney U tests. Univariable and multivariable Cox proportional regression analyses were performed to identify predictors of CSS and OS. Considering that NLR, PLR and SII were confounding factors, we performed three models of multivariable analyses which included one of these factors. *p* < 0.05 was considered to be statistically significant in all analyses.

## 3. Results

### 3.1. Patient Characteristics

Patient characteristics are summarized in Table 1. The median values of NLR, PLR and SII (×10^9^) were 2.2 (quartile: 1.6–3.2), 134.8 (quartile: 99.7–179.9) and 474.2 (quartile: 326.0–782.0), respectively. Seventy-one patients (30%) received NAC. By a pathological examination of resected specimen, 216 patients (91%) and 21 patients (9%) were diagnosed as urothelial carcinoma (UC) and non-UC (squamous cell carcinoma in 10 patients, small cell carcinoma in seven patients and other histopathological type in four patients), respectively. Pathological T stage was pT0-pT2 in 147 patients (62%) and pT3-pT4 in 90 patients (38%). Forty-two patients (18%) had pathological lymph node metastasis.

### 3.2. Comparison of CSS/OS between High SII and Low SII Groups

Figure 1 shows ROC curves of NLR, PLR and SII for CSS. The optimal cutoff values for NLR, PLR and SII (×10^9^) were 2.0, 144 and 438, respectively. The patients were classified into two groups using these cutoff values. Table 2 shows a comparison of patient characteristics between patients with high SII (*n* = 127) and those with low SII (*n* = 110). Distribution of cystectomy approach and urinary diversion differed significantly between the two groups. In addition, the patients with high SII had a higher percentage of non-UC disease (13% vs. 4%, *p* = 0.01) and pathological T stage 3 or more (46% vs. 29%, *p* < 0.01) than those with low SII.

The median follow-up period was 38 months (quartile: 17–64 months), during which 70 patients (30%) and 21 patients (9%) died of bladder cancer and other causes, respectively. Overall, three-year and five-year CSS rates were 74% and 66%, respectively. Figure 2 shows a comparison of CSS between the groups. Patients with high SII had a significantly lower rate of CSS than those with low SII (*p* < 0.01) (Figure 2C). Similar results were obtained for NLR (*p* = 0.01) and PLR (*p* = 0.01) (Figure 2A,B). Three-year and five-year OS rates in overall patients were 69% and 58%, respectively. Figure 3 shows a comparison of OS between groups. Patients with high SII had significantly lower rate of OS than those with low SII (*p* < 0.01) (Figure 3C). Similar results were obtained for NLR (*p* = 0.01) and PLR (*p* = 0.02) (Figure 3A,B).

### 3.3. Predictive Value of Inflammatory Biomarker for CSS/OS

Table 3 shows the results of univariable and multivariable Cox proportional analyses of associations between predictive factors and CSS. In multivariable analysis model 3, pT3-T4 (*p* < 0.01), pN positive (*p* = 0.02) and high SII (*p* = 0.01) were independent significant predictors of poor CSS. Meanwhile, NLR and PLR were not significant (*p* = 0.06 in model 1 and *p* = 0.16 in model 2, respectively).

Table 4 shows the results of univariable and multivariable analyses of associations between various parameters and OS. On multivariable analysis model 3, increasing age (*p* < 0.01), pT3-T4 (*p* < 0.01) and high SII (*p* < 0.01) were independent significant predictors of poor OS. Moreover, in multivariable analysis model 1, high NLR was an independent significant predictor of poor OS (*p* = 0.04) in addition to increasing age (*p* < 0.01) and pT3-T4 (*p* < 0.01). Conversely, in multivariable model 2, high PLR was not an independent significant predictor (*p* = 0.08).

## 4. Discussion

We investigated the impact of preoperative SII on prognosis after RC in patients with bladder cancer. To our knowledge, the usefulness of preoperative SII for predicting long-term survival after RC has not been widely evaluated. Patients with high SII had significantly lower rates of CSS and OS than those with low SII. Moreover, high SII was associated with poor CSS and poor OS independently from patient characteristics such as age and gender, and cancer features including pathological diagnosis, pathological T stage and pathological N stage.

The association between inflammation and cancer has been widely investigated in the last few decades [13]. Immune cells play a major role in the inflammation process, leading to the production of cytokines and chemokines which accelerate cancer growth, neovascularization and metastasis. The complicated balance between inflammation cells and inflammation-related substances could therefore influence the type of peripheral circulating cells. Neutrophils and platelets have been shown to promote the progression of cancer [14]. Neutrophils can induce cell adhesion and tumor seeding through secretion of circulating growth factors [15,16]. Platelets can accelerate epithelial-mesenchymal transition and extravasation of circulating tumor cells [17,18]. On the other hand, lymphocytes can play an antitumoral role through the induction of cytotoxic cell death, the inhibition of tumor cell proliferation and the host’s immune response to tumors [19,20]. Inflammation is, therefore, strongly associated with cancer microenvironment and cancer progression.

Within this context, several biomarkers have been developed based on peripheral circulating blood cell counts. Notably, NLR, which is calculated as neutrophil counts/lymphocyte counts, and PLR, which is calculated as platelet counts/lymphocyte counts, have been widely evaluated as representative prognostic biomarkers in patients with various cancers, including bladder cancer [11,12]. Moreover, a novel inflammation-related biomarker, namely SII, has recently been reported to have greater impact on prognosis than NLR and PLR in patients with malignancies [7,8,9,10]. SII is calculated as platelet count × neutrophil counts/lymphocyte count, which means SII is a combination of NLR and PLR. Excellent predictive power of preoperative SII for prognosis after RC in patients with bladder cancer has been reported. Gorgel et al. retrospectively investigated the prognostic value of the preoperative SII in 191 patients undergoing RC for their muscle invasive bladder cancer [21]. The optimal cutoff value of SII was 843 (×10^9^), determined based on receiver operating characteristic curves for cancer-specific death, similar to our study. In their cohort, CSS and OS rates in patients with high preoperative SII were significantly lower than those with low SII. In addition, high SII was a significant independent predictor of poor CSS in multivariable analysis, while it was not significant for OS. Zhang et al. randomly classified their 209 patients who underwent RC for bladder cancer into a primary cohort (*n* = 139) and a validation cohort (*n* = 70) in a retrospective study [22]. In the primary cohort, x-tile analysis showed that the optimal cutoff value of SII was 507 (×10^9^), and high SII was an independent predictor of poor OS. They developed an OS nomogram using the data on various factors, including preoperative SII in the primary cohort, and showed that the nomogram had high concordance index values for the primary cohort (0.82) and the validation cohort (0.78). However, the association between preoperative SII and CSS was not examined in their study. In the present study, elevated preoperative SII was shown to be independently associated with poor CSS and OS. This is the first known report to show the impact of SII on both CSS and OS.

NLR, PLR and SII were considered to be confounding factors, so we performed three models of multivariable cox progression analyses for CSS and OS. In multivariable analyses for CSS, high SII was a significant predictor of poor CSS (*p* = 0.01), while high NLR and high PLR were not (*p* = 0.06 and *p* = 0.016, respectively). In multivariable analyses for OS, high NLR and high SII were significant predictors of poor OS (*p* = 0.04 and *p* < 0.01, respectively), although high PLR was not (*p* = 0.08). SII could therefore be considered to be a more useful biomarker for predicting prognosis, especially of CSS, than NLR and PLR in bladder cancer patients undergoing RC. This is reasonable considering that peripheral neutrophils and platelets are associated with cancer progression and peripheral lymphocytes are related with cancer suppression. To use SII in future clinical practice, further large-scale studies are required, and identification of the optimal cutoff value of SII for predicting survival after RC in patients with bladder cancer is especially important. The optimal cutoff values of SII for cancer-specific death were 438 (×10^9^) in the present study, and 843 (×10^9^) in the previous study [21]. This difference may be due to the difference in tumor stage between the studies. The rate of pathological T3 or more in our study was 38% (90/237 cases) and lower than that in the previous study (51%, 98/191 cases). Moreover, the rate of pathological N positive in the current study was 18% (42/237 cases) and also lower than that in the previous study (37%, 71/191 patients). Examination of the optimal cutoff value adjusted by tumor stage may therefore be necessary.

Our study has some limitations; it was a retrospective study, and the sample size was relatively small, although it was larger than that of previous studies. In addition, because our institutions are not high-volume centers, the possibility that our lack of experience with RC had an impact on our results cannot be completely denied. Secondly, as described above, the optimal cutoff value of SII for predicting prognosis after RC in patients with bladder cancer has not yet been identified. Thirdly, the timing of laboratory tests was also inconsistent, although the patients in our institutions routinely received preoperative laboratory test within 30 days of RC. Forth, NAC might affect the preoperative NLR, PLR and SII. As shown in the results, 71 patients (30%) received NAC. Although these patients underwent RC after recovery from bone marrow suppression, the ratio of peripheral blood cell counts could be affected by bone marrow suppression or the administration of granulocyte colony stimulating factor accompanied with it. Lastly, we did not examine the association between immunological markers, including various gene scores and ribonucleic acid sequences of tumors, and prognosis after RC. In this study, we focused on the preoperative peripheral blood count-associated factors because they were easy to use clinically and, as a result, showed the prognostic impact of SII in patients who underwent RC. To overcome these limitations, future large-scale prospective studies should be conducted.

## 5. Conclusions

High SII was an independent significant predictor of both poor CSS and poor OS after RC in patients with bladder cancer. SII can be easily calculated from preoperative blood test results. SII may eventually be used in daily clinical practice as a more reliable prognostic predictor than NLR and PLR in patients who undergo RC for bladder cancer.

## Figures and Tables

**Figure 1 diagnostics-11-02194-f001:**
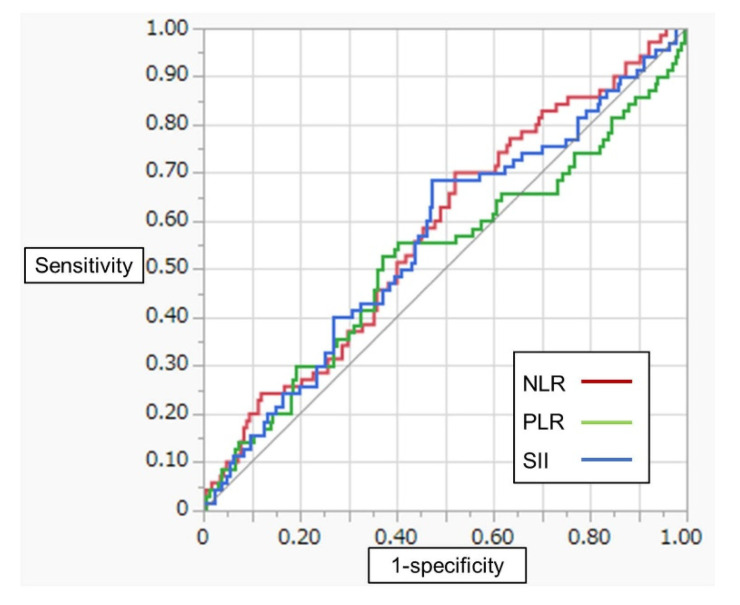
ROC curves of NLR, PLR and SII for CSS.

**Figure 2 diagnostics-11-02194-f002:**
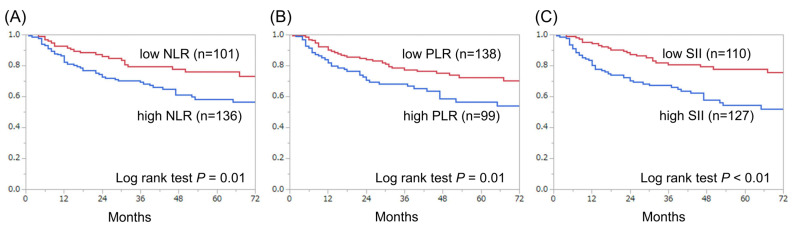
Kaplan–Meier curves for CSS according to preoperative NLR (**A**), PLR (**B**) and SII (**C**).

**Figure 3 diagnostics-11-02194-f003:**
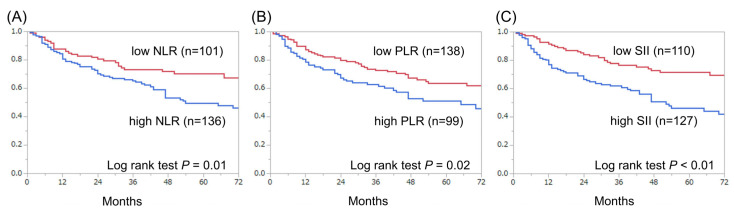
Kaplan–Meier curves for OS according to preoperative NLR (**A**), PLR (**B**) and SII (**C**).

**Table 1 diagnostics-11-02194-t001:** Patient characteristics (*n* = 237).

Age, Years	73 (67–79)
Gender, *n* (%)	
Male	188 (79)
Female	49 (21)
BMI, kg/m^2^	22.3 (19.8–24.4)
ECOG PS 1 or more, *n* (%)	41 (17)
CCI 1 or more, *n* (%)	110 (46)
NLR	2.2 (1.6–3.2)
PLR	134.8 (99.7–179.9)
SII, ×10^9^	474.2 (326.0–782.0)
NAC, *n* (%)	71 (30)
Cystectomy approach, *n* (%)	
Open	196 (83)
Laparoscopic	23 (10)
Robotic	18 (8)
Urinary diversion, *n* (%)	
Cutaneous ureterostomy	93 (39)
Ileal conduit	123 (52)
Neobladder	21 (9)
Pathological diagnosis, *n* (%)	
UC	216 (91)
Non-UC	21 (9)
Pathological T stage, *n* (%)	
pT0-T2	147 (62)
pT3-T4	90 (38)
Lymph node metastasis, *n* (%)	
pN negative	195 (82)
pN positive	42 (18)
CIS concurrent, *n* (%)	
Yes	50 (21)
No	187 (79)
Continuous variables are shown in “median (IQR)” form.	

Abbreviations: BMI = body mass index; ECOG PS = Eastern Cooperative Oncology Group Performance Status; CCI = Charlson Comorbidity Index; NLR = neutrophil-lymphocyte ratio; PLR = platelet-lymphocyate ratio; SII = systemic immune-inflammation index; NAC = neoadjuvant chemotherapy; UC = urothelial carcinoma.

**Table 2 diagnostics-11-02194-t002:** Comparison of patient demographics between patients with high SII and those with low SII.

	High SII Group (*n* = 127)	Low SII Group (*n* = 110)	*p* Value
Age, years	74 (66–79)	73 (68–78)	0.83
Gender, *n* (%)			0.12
Male	96 (76)	92 (84)	
Female	31(24)	18 (16)	
BMI, kg/m^2^	22.1 (19.5–24.0)	22.7 (20.2–24.8)	0.11
ECOG PS 1 or more, *n* (%)	24 (19)	17 (15)	0.48
CCI 1 or more, *n* (%)	63 (50)	47 (43)	0.28
NAC, *n* (%)	41 (32)	30 (27)	0.40
Cystectomy approach, *n* (%)			<0.01
Open	114 (90)	82 (76)	
Laparoscopic	6 (5)	17 (15)	
Robotic	7 (6)	11 (10)	
Urinary diversion, *n* (%)			0.03
Cutaneous ureterostomy	55 (43)	38 (35)	
Ileal conduit	66 (52)	57 (52)	
Neobladder	6 (5)	15 (14)	
Pathological diagnosis, *n* (%)			0.01
UC	110 (87)	106 (96)	
Non-UC	17 (13)	4 (4)	
Pathological T stage, *n* (%)			<0.01
pT0-T2	69 (54)	78 (71)	
pT3-T4	58 (46)	32 (29)	
Lymph node metastasis, *n* (%)			0.39
pN negative	102 (80)	93 (85)	
pN positive	25 (20)	17 (15)	
CIS concurrent, *n* (%)			0.69
Yes	28 (22)	22 (20)	
No	99 (78)	88 (80)	
Continuous variables are shown in “median (IQR)” form.			

Abbreviations: BMI = body mass index; ECOG PS = Eastern Cooperative Oncology Group Performance Status; CCI = Charlson Comorbidity Index; NAC = neoadjuvant chemotherapy; UC = urothelial carcinoma.

**Table 3 diagnostics-11-02194-t003:** Univariable and multivariable analyses of associations between various parameters and cancer-specific survival.

Variable	Univariable Analysis			Multivariable Analysis (Model 1)			Multivariable Analysis (Model 2)			Multivariable Analysis (Model 3)		
	HR	95% CI	*p* Value	HR	95% CI	*p* Value	HR	95% CI	*p* Value	HR	95% CI	*p* Value
Age	1.01	0.98–1.04	0.53	1.02	0.98–1.04	0.26	1.02	0.98–1.05	0.27	1.02	0.99–1.05	0.18
Male (vs. female)	0.88	0.50–1.53	0.65	1.11	0.62–1.97	0.72	1.09	0.61–1.94	0.76	1.17	0.65–2.09	0.58
Non UC (vs. UC)	3.10	1.69–5.69	<0.01	1.59	0.83–3.03	0.16	1.69	0.89–3.21	0.10	1.56	0.82–2.96	0.17
pT3-T4 (vs. pT0-T2)	4.31	2.63–7.05	<0.01	3.33	1.95–5.67	<0.01	3.27	1.90–5.60	<0.01	3.16	1.85–5.41	<0.01
pN positive (vs. pN negative)	2.76	1.66–4.58	<0.01	1.89	1.10–3.23	0.01	1.79	1.04–3.07	0.03	1.84	1.07–3.14	0.02
High NLR (vs. low NLR)	1.89	1.12–3.13	0.01	1.64	0.96–2.79	0.06						
High PLR (vs. low PLR)	1.76	1.10–2.81	0.01				1.41	0.86–2.30	0.16			
High SII (vs. low SII)	2.36	1.42–3.02	<0.01							1.97	1.15–3.34	0.01

Abbreviations: HR = hazard ratio; CI = confidence interval; UC = urothelial carcinoma; NLR = neutrophil-to-lymphocyte ratio; PLR = platelet-to-lymphocyte ratio; SII = systemic immune-inflammation index.

**Table 4 diagnostics-11-02194-t004:** Univariable and multivariable analyses of associations between various parameters and overall survival.

Variable	Univariable Analysis			Multivariable Analysis (Model 1)			Multivariable Analysis (Model 2)			Multivariable Analysis (Model 3)		
	HR	95% CI	*p* Value	HR	95% CI	*p* Value	HR	95% CI	*p* Value	HR	95% CI	*p* Value
Age	1.03	1.00–1.06	0.01	1.04	1.01–1.07	<0.01	1.05	1.01–1.07	<0.01	1.05	1.01–1.07	<0.01
Male (vs. female)	1.05	0.62–1.76	0.84	1.37	0.80–2.33	0.23	1.38	0.81–2.37	0.22	1.50	0.88–2.56	0.13
Non UC (vs. UC)	2.32	1.28–4.19	<0.01	1.32	0.70–2.46	0.39	1.38	0.74–2.58	0.30	1.28	0.68–2.39	0.43
pT3-T4 (vs. pT0-T2)	3.24	2.13–4.93	<0.01	2.82	1.78–4.44	<0.01	2.75	1.73–4.36	<0.01	2.63	1.66–4.15	<0.01
pN positive (vs. pN negative)	2.16	1.35–3.45	<0.01	1.63	0.99–2.67	0.05	1.56	0.94–2.56	0.08	1.63	0.99–2.66	0.05
High NLR (vs. low NLR)	1.73	1.11–2.68	0.01	1.59	1.01–2.51	0.04						
High PLR (vs. low PLR)	1.62	1.06–2.43	0.02				1.46	0.94–2.23	0.08			
High SII (vs. low SII)	2.25	1.44–3.48	<0.01							2.1	1.32–3.30	<0.01

Abbreviations: HR = hazard ratio; CI = confidence interval; UC = urothelial carcinoma; NLR = neutrophil-to-lymphocyte ratio; PLR = platelet-to-lymphocyte ratio; SII = systemic immune-inflammation index.

## Data Availability

The datasets generated during and/or analyzed during the current study are available from the corresponding author on reasonable request.

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
