# Peer review of "Usefulness of Preoperative High Systemic Immune-Inflammation Index as a Prognostic Biomarker in Patients Who Undergo Radical Cystectomy for Bladder Cancer: Multicenter Analysis"

_diagnostics, 2021, doi:10.3390/diagnostics11122194_

Round 1
Reviewer 1 Report
Dear Authors,
your paper is interesting, nevertheless some points might be improved
1) From 2009 to 2018 in 3 centers the total number of RC performed was 239, which make in that time frame less than 20 RC for each centre yearly. This might impact on results, as they should be considered low volume centers. This should be stated in limits
2) In results, please report ROC curves and cutoff values, at least as an additional figure. Visualization of the curve it's fundamental to evaluate its reliability, and thus relevance of your results
Author Response
Dear Authors,
your paper is interesting, nevertheless some points might be improved
1) From 2009 to 2018 in 3 centers the total number of RC performed was 239, which make in that time frame less than 20 RC for each centre yearly. This might impact on results, as they should be considered low volume centers. This should be stated in limits
Thank you for your important suggestion. We added the following sentence to the limitation section.
In addition, because our institutions are not high-volume centers, the possibility that our lack of experience with RC had an impact on our results cannot be completely denied. (page 7, lines 216-218)
2) In results, please report ROC curves and cutoff values, at least as an additional figure. Visualization of the curve it's fundamental to evaluate its reliability, and thus relevance of your results
We appreciate your valuable comment. We added the figure which shows ROC curves and the following sentence to the Results section. Accordingly, Figure 1 and Figure 2 were revised to Figure 2 and Figure 3, respectively.
Fig 1. shows ROC curves of NLR, PLR and SII for CSS. (page 4, lines 110)
Reviewer 2 Report
In the era of immunotherapy and NGS of bladder cancer this study is outdated, overly simplistic and of limited value. There are many more immunologic markers of outcome after radical cystectomy including PDL-1, TMB, interferon gamma and TGF beta gene scores, RNA seq of tumors, etc. This is outdated and of no interest.
Author Response
In the era of immunotherapy and NGS of bladder cancer this study is outdated, overly simplistic and of limited value. There are many more immunologic markers of outcome after radical cystectomy including PDL-1, TMB, interferon gamma and TGF beta gene scores, RNA seq of tumors, etc. This is outdated and of no interest.
We appreciate your important comments.
As you pointed out, this is the era of immunotherapy and next-generation sequencing has been widely used in a lot of studies about various cancers. However, radical cystectomy remains the gold standard treatment of muscle invasive bladder cancer. Although various immunologic markers could be also useful for predicting prognosis after radical cystectomy in patients with bladder cancer, we believe that systemic immune inflammation index has the merit of being easy to use clinically and our results. We added the following sentence to the limitation section.
In the future, we would like to try to examine the association between the immunologic markers and prognosis with bladder cancer.
Lastly, we did not examine the association between immunological markers, including various gene scores and ribonucleic acid sequences of tumors, and prognosis after RC. In this study, we focused on the preoperative peripheral blood count-associated factors because they were easy to use clinically and, as a result, showed the prognostic impact of SII in patients who underwent RC. (page 226-230)
Reviewer 3 Report
The study design is well structured and results are well presented. The paper is well written and opens the door to further investigate the role of immune-inflammation indexes as prognostic biomarkers for patients undergoing radical cystectomy for bladder cancer.
The study design, however, raises the following concern: Neoadjuvant chemotherapy may represent a bias. Patients receiving platinum based chemo, may necessitate of bone marrow growth factors. The final neutrophil / lymhocyte count may be affected by the chemo as such, and/or by the growth factors (when administered). How have you addressed this bias? Please comment.
Author Response
The study design is well structured and results are well presented. The paper is well written and opens the door to further investigate the role of immune-inflammation indexes as prognostic biomarkers for patients undergoing radical cystectomy for bladder cancer.
We appreciate your kind comments. Thank you very much.
The study design, however, raises the following concern: Neoadjuvant chemotherapy may represent a bias. Patients receiving platinum based chemo, may necessitate of bone marrow growth factors. The final neutrophil / lymhocyte count may be affected by the chemo as such, and/or by the growth factors (when administered). How have you addressed this bias? Please comment.
We appreciate your important comments.
Seventy-one patients (30%) in our cohort received neoadjuvant chemotherapy. As you pointed out, although these patients underwent radical cystectomy after recovery from bone marrow suppression, the ratio of peripheral blood cell counts could be affected by bone marrow suppression or the administration of granulocyte colony stimulating factor drugs. Therefore, we added the following sentences to the limitation section.
Forth, NAC might affect the preoperative NLR, PLR and SII. As shown in the results, 71 patients (30%) received NAC. Although these patients underwent RC after recovery from bone marrow suppression, the ratio of peripheral blood cell counts could be affected by bone marrow suppression or the administration of granulocyte colony stimulating factor accompanied with it. (page 8, lines 222-226)